# Effect of Y$_2$O$_3$ Concentration on the Surface and Bulk Crystallization of Multicomponent Silicate Glasses

Akram Beniaiche [1], Aitana Tamayo [2], Nabil Belkhir [1], Fausto Rubio [2], Abdellah Chorfa [1] and Juan Rubio [1,*]

1   Laboratory of Applied Optics, Institute of Optics and Precision Mechanics, Ferhat Abbas University, Setif 19000, Algeria

2   Institute of Ceramic and Glass, Consejo Superior de Investigaciones Científicas (CSIC), 28049 Madrid, Spain; aitanath@icv.csic.es (A.T.); frubio@icv.csic.es (F.R.)

*   Correspondence: jrubio@icv.csic.es; Tel.: +34-91-735-5840

**Abstract:** Multicomponent silicate glasses are crystallized by Y$_2$O$_3$ addition. Depending on the Y$_2$O$_3$ concentration, different crystalline phases evolve. In the absence of Y$_2$O$_3$, a multicomponent glass crystallizes as ZnSnO$_3$, while with the addition of just 3% of this oxide, ZnSnO$_3$ no longer crystallizes and ZrSiO$_4$ appears instead. Different yttrium silicate crystals are formed in all glasses containing Y$_2$O$_3$, but, while α-Y$_2$Si$_2$O$_7$ and β-Y$_2$Si$_2$O$_7$ are favored at low Y$_2$O$_3$ concentrations, the γ-Y$_2$Si$_2$O$_7$ and y-Y$_2$Si$_2$O$_7$ phases are favored at the maximum Y$_2$O$_3$ content. At a 12% Y$_2$O$_3$ concentration, barium and calcium silicate crystalline phases also evolve. Interestingly, the crystalline phases appearing on the surface of the material present different microstructures compared to crystals developed in the bulk. While the crystallized surface presents a tabular-shape type, crystallization in the bulk is of a prismatic type at low Y$_2$O$_3$ concentrations and of a globular (spherical) type at higher concentrations. The main crystal size ranges between 0.85 and 0.75 micrometers, but most of the crystals coalesce to form larger superstructures depending on the Y$_2$O$_3$ concentrations.

**Keywords:** crystallization; Y$_2$O$_3$; multicomponent glasses; microstructure; surface and bulk crystals





## 1. Introduction

Global warming is leading to an increase in the use of air-conditioning systems in cities with a high population, which also leads to an increase in the environmental temperature due to the abuse of these systems. This is a continuous process that leads to ceaselessly incremental temperature values in cities, an effect that is known as the urban heat island (UHI) effect. To mitigate the UHI effect, several strategies have been proposed and tested in the best cases [1–4]. Two of these strategies consist of the use of white glass enamels and/or glass beads [5]. White glass enamels can display high reflectivity to solar radiation due to the presence of micrometric crystals of high refractive indexes within their structure [6,7], whereas in the case of glass beads, the high retro-reflectivity effect is accomplished due to their spherical shape, leading to the reflection of most solar radiation and, thus, avoiding its transmission to the inner side of buildings [8,9]. In this case, glass beads are small transparent glass spheres deposited and fixed over a white substrate, with these being mainly constituted of zirconium silicate (ZrSiO$_4$) because of its high refractive index (n$_d$ = 1.94–1.96). Nevertheless, the use of bead compositions such as barium titanate (n$_d$ = 2.43) has also been reported, although in this case, the colorimetric appearance of the substrate is changed [9].

White substrates such as glass enamels, in which different forms of ZrSiO$_4$ crystallize, have been widely used for a long time in the ceramic tile industrial sector [10], but the rising price of zirconium oxide (ZrO$_2$) has led to the development of new materials in which other crystalline phases, such as gahnite, also appear (ZnAl$_2$O$_4$; n$_d$ = 1.79–1.80) [11]. At the same time, glasses with high refractive indexes (n$_d$ = 1.59–1.62) might also be used [12], although for those with very high refractive indexes (n$_d$ = 2.26–2.33), it is also

necessary to employ some other expensive oxides such as $La_2O_3$ and $Nb_2O_5$ [13]. There are other oxides that are less expensive ($SnO_2$, ZnO, BaO, CaO, MgO, etc.), which can be used for preparing high-refractive-index glasses and are also used as raw materials for the manufacturing of glass enamels [14–16]. In the case of $Y_2O_3$, it can serve as a material for the preparation of high-refractive-index glasses as a network former or as a network modifier, depending on its concentration [17]. With these premises, in the literature, there are reports on several glass compositions containing $Y_2O_3$ in combination with some other oxides, such as $Li_2O$ [18], $Al_2O_3$ [19], MgO [20], ZnO [21], and $ZrO_2$ [22]. The incorporation of different oxides ($Y_2O_3$, $SnO_2$, ZnO, BaO, etc.) into the base of a silicoborate glass generally leads to the obtainment of high-refractive-index glasses that can be used in optical applications [18,23–25]. However, when these glasses are heat-treated at high temperatures, they are prone to crystallizing. The effect of $ZrO_2$ and $ZrSiO_4$ has been widely studied for their incorporation in multicomponent glasses and glass enamels in the ceramic tile sector. However, this effect has not been analyzed for $Y_2O_3$, even though this oxide has interesting properties related to its high refractive index and the possibility to produce glass enamels of high whiteness. The objective of this work is to study the effect of $Y_2O_3$ concentration on the crystallization behavior of a multicomponent oxide silicoaluminate glass containing $Li_2O_3$, $Na_2O$, $K_2O$, MgO, CaO, BaO, ZnO, $ZrO_2$, $SnO_2$, and $B_2O_3$. These crystallized materials can be used as glass enamels for high-reflecting surfaces in ceramic tiles for roofs and façades, home ovens, fillers for white paints, etc.

## 2. Experimental Section

### 2.1. Glass Preparation and Crystallization

All glasses prepared in this work were obtained from the following raw materials: silica quartz, $H_3BO_3$, $Al_2O_3$, $Na_2CO_3$, $K_2CO_3$, $MgCO_3$, $CaCO_3$, $BaCO_3$, $ZrO_2$, ZnO, $SnO_2$, and $Y_2O_3$, with all of them being of 99.9% purity. The appropriate amounts of these raw materials were thoroughly mixed for 2 h and then heat-treated in a Pt crucible. The thermal cycle to which these raw materials were subjected was a first pre-treatment at 900 °C for 1 h to fully oxide carbonates and, subsequently, the temperature was raised to 1500 °C and maintained for another 2 h to allow melting and homogenization. The melted glasses were then poured on metallic plates and left to cool to room temperature. Finally, the obtained glasses were annealed at 550 °C to remove internal stresses. A fraction of the glass was cut into small pieces sized $2 \times 2 \times 0.5$ cm, and the remaining materials were agate-milled to obtain particles below 50 µm.

The crystallization schedule was in accordance with the differential thermal analysis (DTA) results, as will be described in the Results section. All DTA curves presented exothermic peaks that were assigned to the formation of crystals inside the glass matrix. The crystallization procedure consisted of heating at a rate of 5 °C/min, followed by holding times of 30 min and then cooling at a rate of 5 °C/min. After this process, all glasses were stored for further characterization. Figure 1 shows an image of the as-prepared and crystallized glasses, and the white color obtained after crystallization can be observed due to the formation of different crystals, as will be described below.

### 2.2. Characterization

The chemical composition of the obtained glasses (Table 1) was analyzed by X-ray fluorescence (Philips, MagiX, Eindhoven, The Netherlands). The materials were characterized by X-Ray diffraction (XRD) using a Bruker D8 Advance fitted with a Cu $K_{\alpha 1}$ (1.540598 Å) anode and working at 40 kV. All the measurements were carried out in the 2θ range of 5–70°. Raman spectra were obtained with a spectrometer (Renishaw InVia, Gloucestershire, UK) using a 514 nm laser in the region from 100 to 1500 $cm^{-1}$ and for each spectrum 10 scans were collected (background subtraction was performed in all the cases by assuming a 6-order polygonal spline line). UV–Vis–NIR reflectance was obtained with a spectrophotometer (PerkinElmer, Lambda 950, Waltham, MA, USA) equipped with an integrating sphere in the 280–2500 nm spectral range. The microstructures of the crystalline

phases developed in the glass matrix were observed by using a field emission scanning electron microscope (FE-SEM, Hitachi S-4700, Chiyoda, Tokyo, Japan) on the fractured surface attacked by hydrofluoric acid (HF, 5%) for 5 s, rinsed and dried at 100 °C for 1 h, and finally coated with Au. Energy dispersive analysis (EDS) on fracture surfaces was carried out by using a Noran X-ray instrument coupled to the FE-SEM (Thermo Fisher Scientific, Waltham, MA, USA). Each analysis was carried out for 10 min at 20 kV.

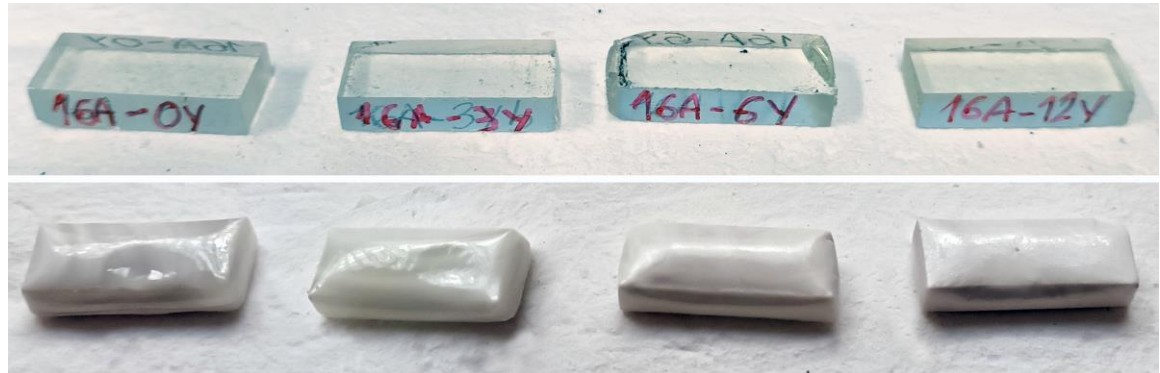

**Figure 1.** Images of the as-prepared (**top**) and crystallized (**bottom**) glasses.

**Table 1.** Chemical compositions (wt %) of the prepared glasses.

| Component | AK-0Y | AK-3Y | AK-5Y | AK-12Y |
|---|---|---|---|---|
| $SiO_2$ | 50.40 | 48.60 | 44.90 | 38.60 |
| $Al_2O_3$ | 14.10 | 14.20 | 14.45 | 14.80 |
| $Na_2O$ | 0.86 | 0.86 | 0.86 | 1.50 |
| $K_2O$ | 1.04 | 0.88 | 0.97 | 1.10 |
| MgO | 6.50 | 6.40 | 5.06 | 6.75 |
| CaO | 9.25 | 9.69 | 9.86 | 9.90 |
| BaO | 5.10 | 5.00 | 5.06 | 3.90 |
| ZnO | 4.26 | 3.82 | 3.53 | 3.03 |
| SnO | 3.81 | 3.65 | 3.86 | 4.16 |
| $ZrO_2$ | 4.00 | 3.72 | 3.78 | 3.50 |
| $Y_2O_3$ | 0 | 2.85 | 5.06 | 11.90 |
| Refractive index | 1.651 | 1.653 | 1.654 | 1.655 |

## 3. Results

### 3.1. Glass Characterization

The obtained glasses were all transparent and with very little brown coloration. Figure 2 shows the XRD diffractograms of the prepared glasses where there is a noticeable absence of any crystallization peak. Additionally, a broad halo appears in the range $2\theta = 17–40°$ attributed to the silicoaluminate glass structure that indicates the absence of any long-range translational order [26]. As the amount of $Y_2O_3$ increases in the glass composition, a slight displacement to higher $2\theta$ is observed. This result indicates significant modifications that this oxide is producing in the glass structure. $Y_2O_3$ is known to act as a network former for concentrations lower than 5% and as a network modifier for higher concentrations [20]. Therefore, it can be inferred that the XRD patterns of the AK-5Y and AK-12Y in Figure 2 show the highest displacement to a higher $2\theta$ range. Additionally, a shift of the main band of the FTIR spectra is also observed (Figure 3) when the $Y_2O_3$ content is increased. In the FTIR of all the glasses, the three characteristic bands observed at approximately 1000 cm$^{-1}$, 725 cm$^{-1}$, and 460 cm$^{-1}$ can be assigned to Si-O-X bonds where X equals Al, Y, Zr, Sn [22,27,28]. It should be noted that in these materials, there is also the possibility of bonding to Zn [16], although due to the low concentration of this element, it is likely acting as network modifier [29].

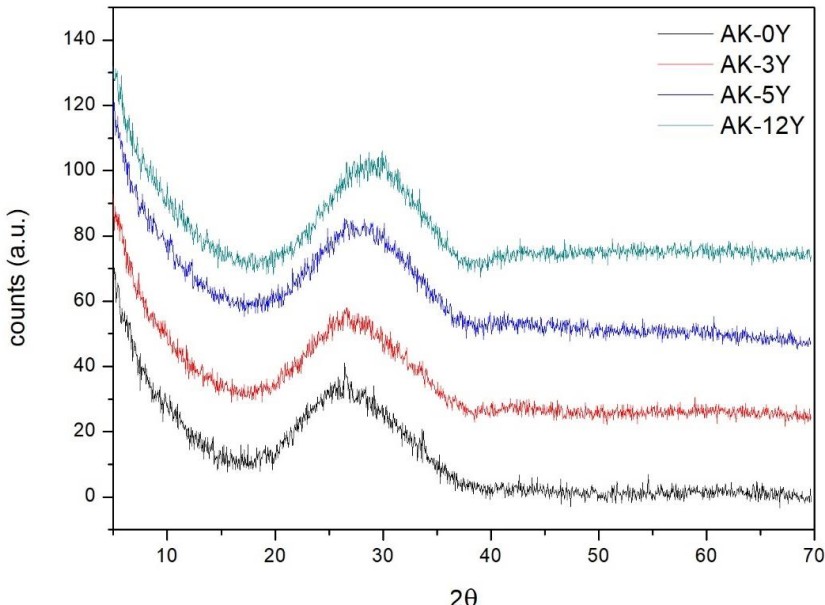

**Figure 2.** XRD of the prepared glasses.

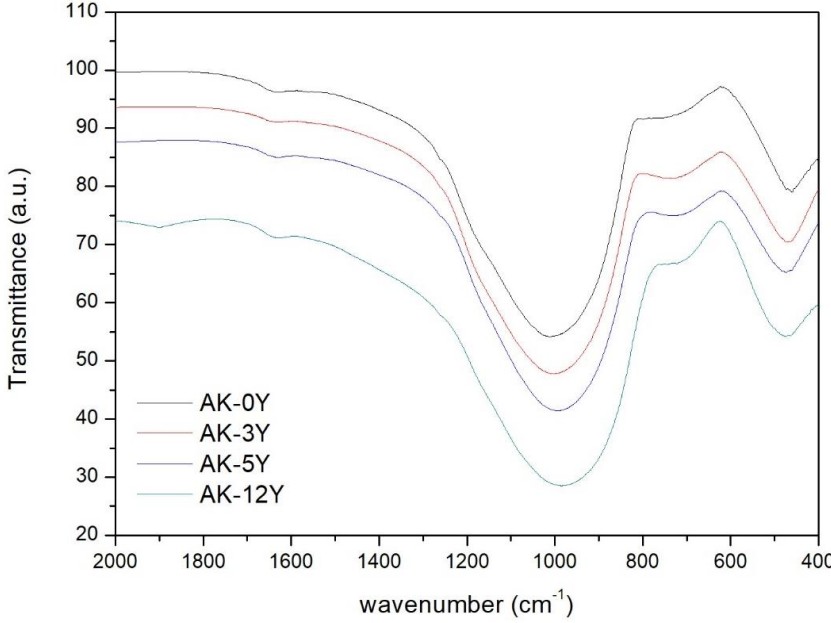

**Figure 3.** FTIR spectra of the prepared glasses.

Regarding the main band in the FTIR spectra (asymmetric stretching of the Si-O-X bond), it is centered at 1009 cm$^{-1}$ in the glass with no Y$_2$O$_3$. As the Y$_2$O$_3$ concentration increases from 0 to 3, 5, and 12%, this band shifts towards 1000, 988, and 966 cm$^{-1}$, respectively. This shift suggests that the Y$_2$O$_3$ is effectively contributing to the change likely through the contribution of Si-O-Y bonds where the Y$_2$O$_3$ acts as a network former [30]. The symmetric stretching of the Si-O-X bond is observed in the FTIR at about 725 cm$^{-1}$ and this band is also affected by the same displacement to lower frequencies. However, it is noticed that the O-Si-O bending appearing at 460 cm$^{-1}$ does not change in position, indicating that only the stretching modes are influenced by the Y$_2$O$_3$.

The Raman spectra in Figure 4 show two high-intensity and broad bands around 990 cm$^{-1}$ and 510 cm$^{-1}$, which are attributed to the Si-O-X bonds. Depending on the composition, there are additional medium-intensity bands centered around 1120 cm$^{-1}$, 750 cm$^{-1}$, and 360 cm$^{-1}$, as well as three small bands or shoulders at 1180 cm$^{-1}$, 680 cm$^{-1}$,

and 610 cm$^{-1}$. The wide bands located at 980 and 1120 cm$^{-1}$ are assigned to the Si-O-X bonds with X = Al, Y, Zr, Sn [31–34] and to the presence of Q$^n$ units (n = 0, 1, 2, 3, 4) where Q represents the SiO$_4$ tetrahedra and n the number of bridging oxygens (BOs). These Q$^n$ units are formed after the BOs are broken by network modifier elements (Na, K, Ca, etc.) to form non-bridging oxygens (NBOs) [32,33].

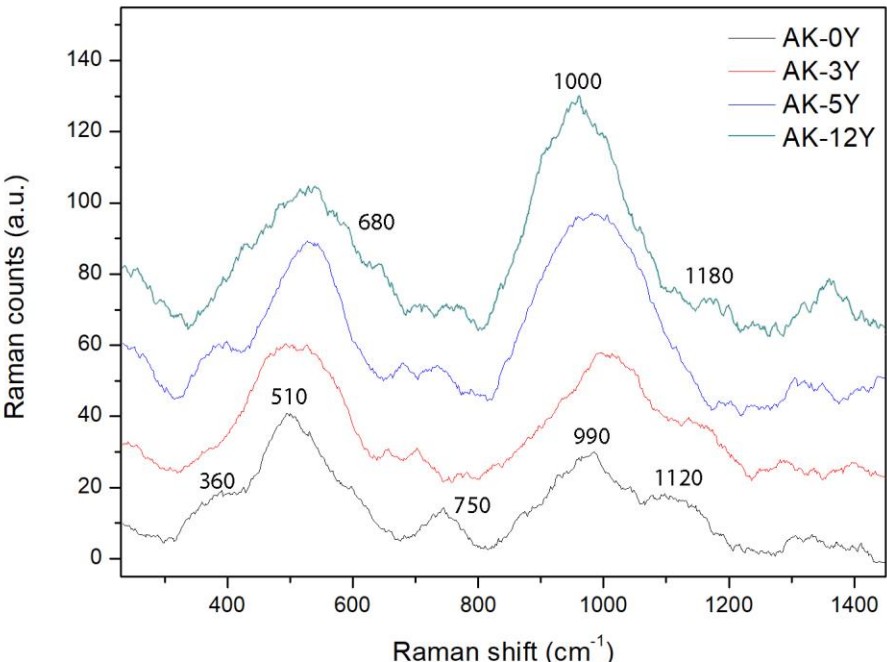

**Figure 4.** Raman spectra of the prepared glasses.

The medium-intensity band centered at 1120 cm$^{-1}$ must be assigned to the Q$^4$ units, representing a three-dimensional SiO$_2$ network. On the other hand, the band at 990 cm$^{-1}$ should be attributed to either the Q$^3$ (Si$_2$O$_5{}^{2-}$), Q$^2$ (SiO$_3{}^{2-}$) or Q$^1$ (Si$_2$O$_7{}^6$) units. This indicates the appearance of NBOs in the glass composition due to the presence of network modifier elements [35]. In the Raman spectra, it is also noticed that the band at 1120 cm$^{-1}$ is only present in the AK-0Y and AK-3Y glasses, disappearing when the Y$_2$O$_3$ composition is 5% and 12% (glasses AK-5Y and AK-12Y). This observation is attributed to the disappearance of the SiO$_4$ units to form new NBOs that, together with the shift of the band centered at 1000 cm$^{-1}$ towards lower frequencies and it increases in intensity as the Y$_2$O$_3$ concentration increases, corroborates the effect of Y$_2$O$_3$ as a network modifier at the highest concentrations and the formation of large amounts of NBO [34].

Figure 5 displays the DTA curves of the prepared glasses, indicating several temperature phenomena such as the glass transition temperature (T$_g$) and the exothermic peak of crystallization (T$_p$) along with its corresponding temperature onset (T$_x$) and the melting temperature (T$_m$). It can be observed that for the AK-12Y sample, two endothermic T$_m$ peaks appear, indicating at least two phases that melt at very close temperatures. As shown below, this glass has several crystalline phases that may have different melting temperatures. The actual values of these characteristic temperatures are provided in Table 2. The glass transition temperature corresponds to the temperature at which the atoms suffer a structural relaxation, and the structural units rearrange due to the high viscosity of the solid glass [26]. Based on these temperatures, several crystallization parameters can be calculated such as the glass stability ($K_H$, Equation (1)), the glass-forming ability ($\beta$, Equation (2)), and tendency to crystallize ($T_C$, Equation (3)) [36]:

$$K_H = \frac{(T_x - T_g)}{(T_{m1} - T_x)} \tag{1}$$

$$\beta = \frac{T_x T_g}{(T_{m1} - T_x)^2} \tag{2}$$

$$T_c = T_x - T_g \tag{3}$$

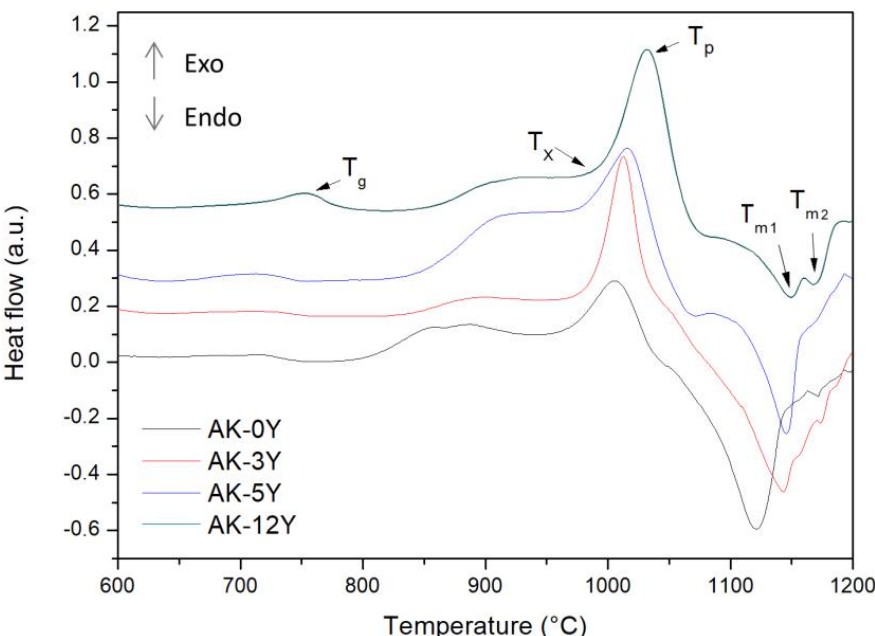

**Figure 5.** DTA curves of the prepared glasses.

**Table 2.** Characteristic temperatures ($T_g$, $T_x$, $T_m$) and glass stability parameters.

|  | AK-0Y | AK-3Y | AK-5Y | AK-12Y |
|---|---|---|---|---|
| $T_g$ (±6 °C) | 721 | 725 | 735 | 759 |
| $T_x$ (±8 °C) | 959 | 963 | 969 | 972 |
| $T_{m1}$ (±1 °C) | 1121 | 1145 | 1147 | 1150 |
| $K_H$ | 1.47 | 1.31 | 1.31 | 1.13 |
| $\beta$ | 26.35 | 21.08 | 22.48 | 20.87 |
| $T_C$ (°C) | 238 | 238 | 234 | 213 |

Among these parameters, $K_H$ and $T_C$ represent the resistance against crystallization while the parameter $\beta$ represents the relative ease of obtaining a glass after cooling the melted composition. Table 2 provides the corresponding values for the characteristic temperatures and the calculated stability parameters. It is observed that $T_g$, $T_x$, and $T_p$ increase with the $Y_2O_3$ concentration, indicating that this oxide is being incorporated into the glass network as its concentration is raised. However, the calculated parameters $K_H$, $\beta$, and $T_C$ show a continuous decrease with the concentration of $Y_2O_3$, a result that is interpreted as a decrease in the stability against crystallization as the $Y_2O_3$ amount is raised. According to the literature, it has been demonstrated that the incorporation of low concentrations (below 3%) of $Y_2O_3$ into different glass compositions produces an increase in the stability of the obtained glasses because the $Y^{3+}$ ions become tetrahedrally coordinated and act as a network former [20].

### 3.2. Crystallization Process

The DTA results presented in Figure 5 and the data of Table 2 were used for the subsequent treatments applied in all glasses to develop the crystal phases. According to the position of the exothermic peaks, the glasses were heat-treated at 1040 °C for 10 and 30 minutes. In all cases the transparent glasses were transformed into white glass-

ceramic materials due to the crystallization process and the formation of the different crystalline phases. No significant differences were observed by varying the dwell time at the maximum temperature of crystallization. Therefore, in this work, we only present the results corresponding to 30-minute dwell time since crystallization was enhanced. Figure 6 shows the high-intensity XRD reflections corresponding to the crystallized phases. An interesting change occurs when just 3% $Y_2O_3$ is added to the glass composition since the crystallization peaks of the AK-0Y sample completely disappear and are substituted by several other phases that are maintained at the other different concentrations. This indicates the powerful effect of the $Y_2O_3$ on the crystallization process.

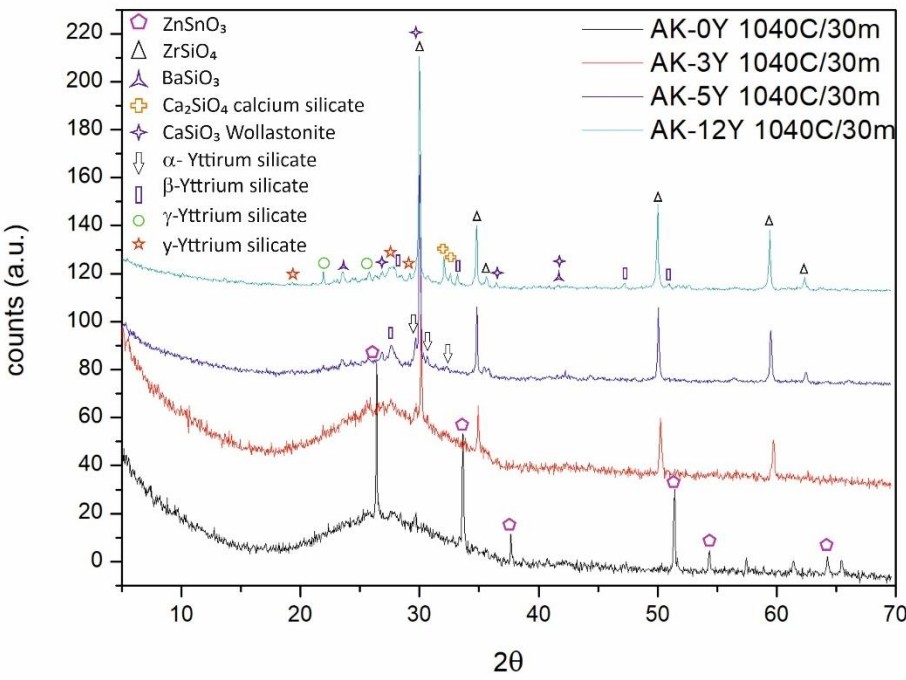

**Figure 6.** XRD patterns of the crystallized samples.

The XRD peaks of the AK-0Y sample correspond to the reflections of $ZnSnO_3$ crystals (JCPDS-028-1486), a phase that disappeared with the sole incorporation of the $Y_2O_3$, as mentioned above. In the AK-3Y, AK-5Y, and AK-12Y samples, zirconium silicate ($ZrSiO_4$, JCPDS-003-0655) is encountered as the main crystalline phase growing together with other minor crystalline phases such as barium silicate ($BaSiO_3$, JCPDS-026-0179), calcium silicate ($Ca_2SiO_4$, JCPDS-029-0369), wollastonite ($CaSiO_3$, JCPDS 01-084-0655), and several yttrium silicates ($\alpha$-$Y_2Si_2O_7$ JCPDS-021-1457, $\beta$-$Y_2Si_2O_7$ PCPDS-021-1454, $\gamma$-$Y_2Si_2O_7$ JCPDS-042-0167, y-$Y_2Si_2O_7$ JCPDS-048-1623). The crystallization complexity increases as the $Y_2O_3$ concentration rises. The sample with 3% $Y_2O_3$ mainly exhibits the formation of the $ZrSiO_4$ phase and most of the other minority crystals are present in very small amounts. Increasing the concentration of $Y_2O_3$ to 5% leads to the increased appearance of these other minority phases. A close analysis of the XRD patterns reveals that both the $\alpha$-$Y_2Si_2O_7$ and $\gamma$-$Y_2Si_2O_7$ crystalline phases are still present in the samples containing 3% and 5% $Y_2O_3$ while these phases decrease in the AK-12Y sample, and the y-$Y_2Si_2O_7$ crystallizes instead, acquiring a higher intensity than that of both $\alpha$- and $\gamma$-yttrium silicate phases. Additionally, in all the XRD patterns there is a broad halo attributed to the glassy (or non-crystallized) phase which decreases in intensity with the $Y_2O_3$ concentration. This effect is due to the fact that this oxide leads to an increase in the tendency to form crystals inside the glass. Vomacka and Babushkin [22] already demonstrated that the presence of $ZrO_2$ and $Y_2O_3$ in a glass material increases the rate of nucleation and crystal formation, exerting a positive influence on glass crystallization.

The glassy phase mentioned above can be better analyzed through Raman spectroscopy carried out on the crystallized samples (Figure 7). The spectra of the samples AK-0Y and AK-3Y are similar to the ones presented in Figure 4, typical of a glassy material (no crystals were detected). In the spectra of the samples containing up to 5% $Y_2O_3$, the main band is broad and located at about 1000 cm$^{-1}$, shifting to 1019 and to 1026 cm$^{-1}$ for the AK-3Y and AK-5Y samples, respectively. This relative displacement of the band indicates that some of the $Y^{3+}$ cations may cease to act as network modifiers and become network formers. In the spectrum of the sample containing 5% $Y_2O_3$, small bands appear at about 510, 880, 960, and 1010 cm$^{-1}$, which become well-defined peaks in the spectrum of the sample AK-12Y. At the highest $Y_2O_3$ concentration, several other bands are also noticed at 330 cm$^{-1}$, 390 cm$^{-1}$, 450 cm$^{-1}$, and 660 cm$^{-1}$.

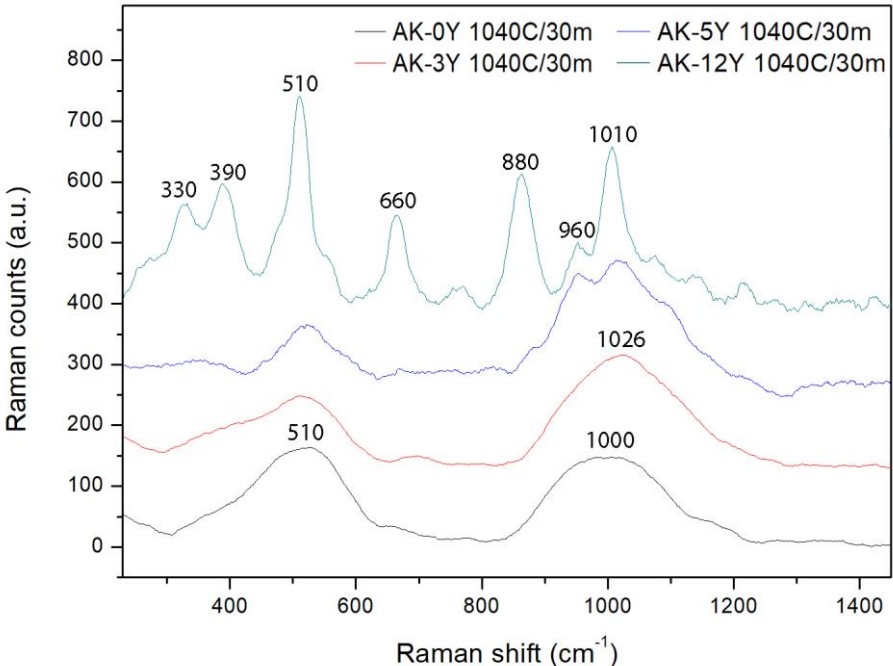

**Figure 7.** Raman spectra of the crystallized samples.

$ZrSiO_4$ has characteristic Raman bands at 1010, 450, and 330 cm$^{-1}$ [37] whereas $Ca_2SiO_4$ and $CaSiO_3$ show their fingerprint bands at 1010, 960, 860, 660, and 330 cm$^{-1}$ [38]. On the other side, different yttrium silicates provide Raman spectra composed of several bands, with the most important ones located at about 400 cm$^{-1}$, 880 cm$^{-1}$, and 905 cm$^{-1}$ [39,40]. Since most of these bands are very close to each other, the Raman spectra of the polycrystalline materials result in an overlapping of the spectra of the different constituents, leading to wide bands that are difficult to uniquely assign. Nevertheless, it should be emphasized that all the Raman spectra show the presence of a glass phase intricated with the crystallized ones, indicating that the developed crystals are widespread all over the glass matrix as described below.

The microstructures of the crystallized glasses as observed by FE-SEM are shown in Figures 8 and 9. The micrographs show that the samples present two zones of crystallization, one starting on its surface (Figure 8) and the other appearing in the bulk (Figure 9). On the outer surface of the samples, a rough crystallized fracture is observed while the microstructure of the bulk fracture surface appears smooth with widespread crystals. The length of this rough crystallized fracture in the outer surface varies from 42 ± 4 for 0% $Y_2O_3$ to 53 ± 5, 560 ± 12, and 592 ± 12 for increasing amounts of this oxide. This observation suggests that the surface crystallization evolves inwards towards the bulk of the sample. Similar results have been reported by several other authors in different materials, such as Zheng et al., who studied yttrium aluminosilicate glasses [34], Wang et al. who

described the same observations in zinc borosilicate glasses [41], and Wisniewski et al. for MgO/Y$_2$O$_3$/SiO$_2$/Al$_2$O$_3$/ZrO$_2$ glasses [42]. According to these observations, it can be deduced that for Y$_2$O$_3$ concentrations below 6%, the Y$_2$O$_3$ does not act as a crystallization promoter and the crystallization is mainly driven by the glass constituents. However, for concentrations above 6%, this oxide promotes the crystallization of the different phases.

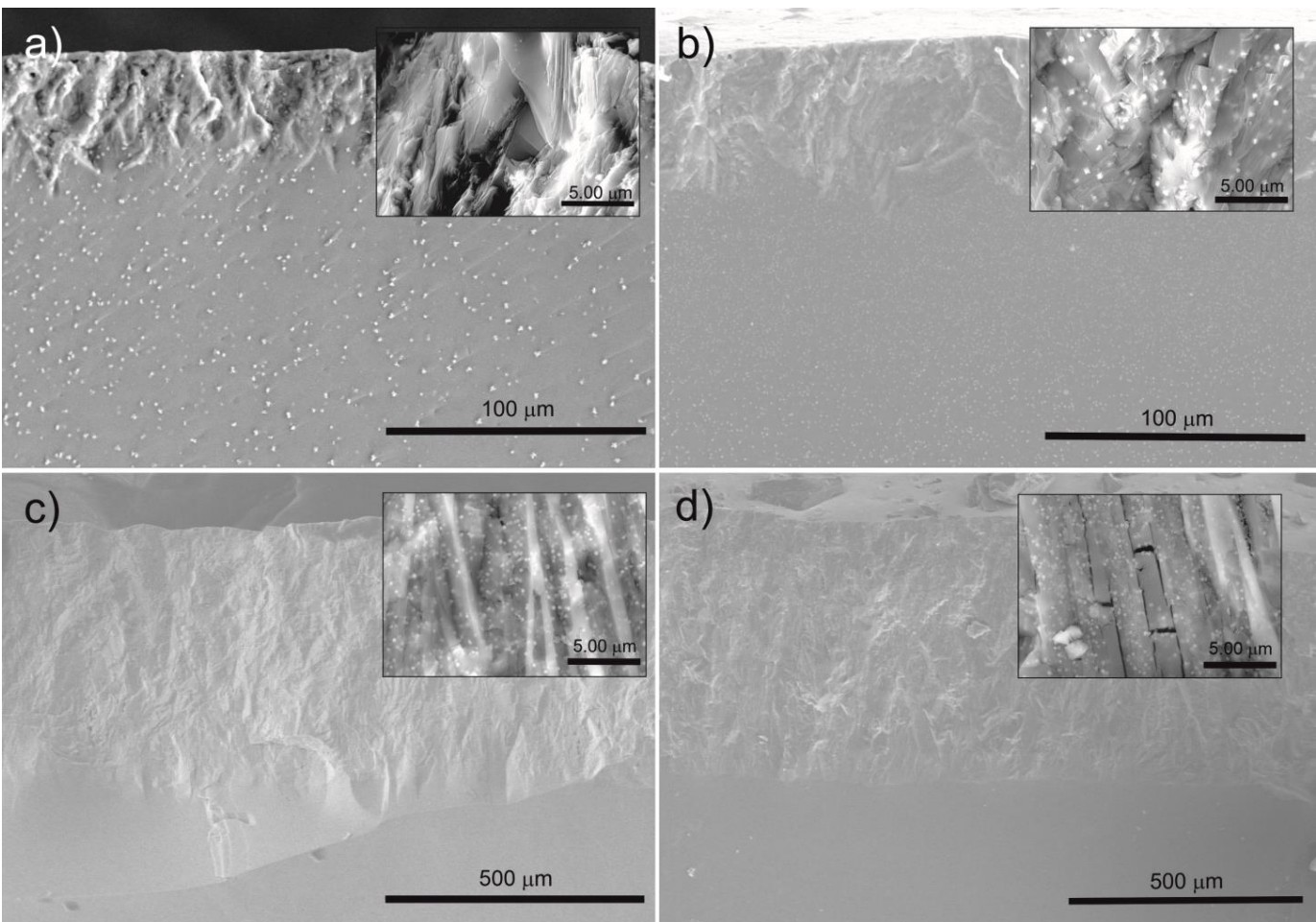

**Figure 8.** Microstructures of the fractures of crystallized glasses at the surface of the samples. (**a**) AK-0Y. (**b**) AK-3Y. (**c**) AK-5Y. (**d**) AK-12Y.

It should be emphasized that this surface crystallization effect is not exclusive to this multicomponent glass but has also been found in several other Y$_2$O$_3$-containing materials [19]. In these micrographs, the surface-crystallized zone also changes with the Y$_2$O$_3$ concentration in the sense that, in the absence of Y$_2$O$_3$, the crystals do not maintain any specific shape whereas tabular-shaped crystals begin to appear clearly for the 3% Y$_2$O$_3$ concentration and dominate the microstructure of the AK-12Y sample. In addition, in the presence of the Y$_2$O$_3$ oxide, either inside or over the prismatic crystals, there also appear small crystals in the shape of prisms and spheres (globular shape). The amount of these small crystals increases with the Y$_2$O$_3$ content and their size decreases as well.

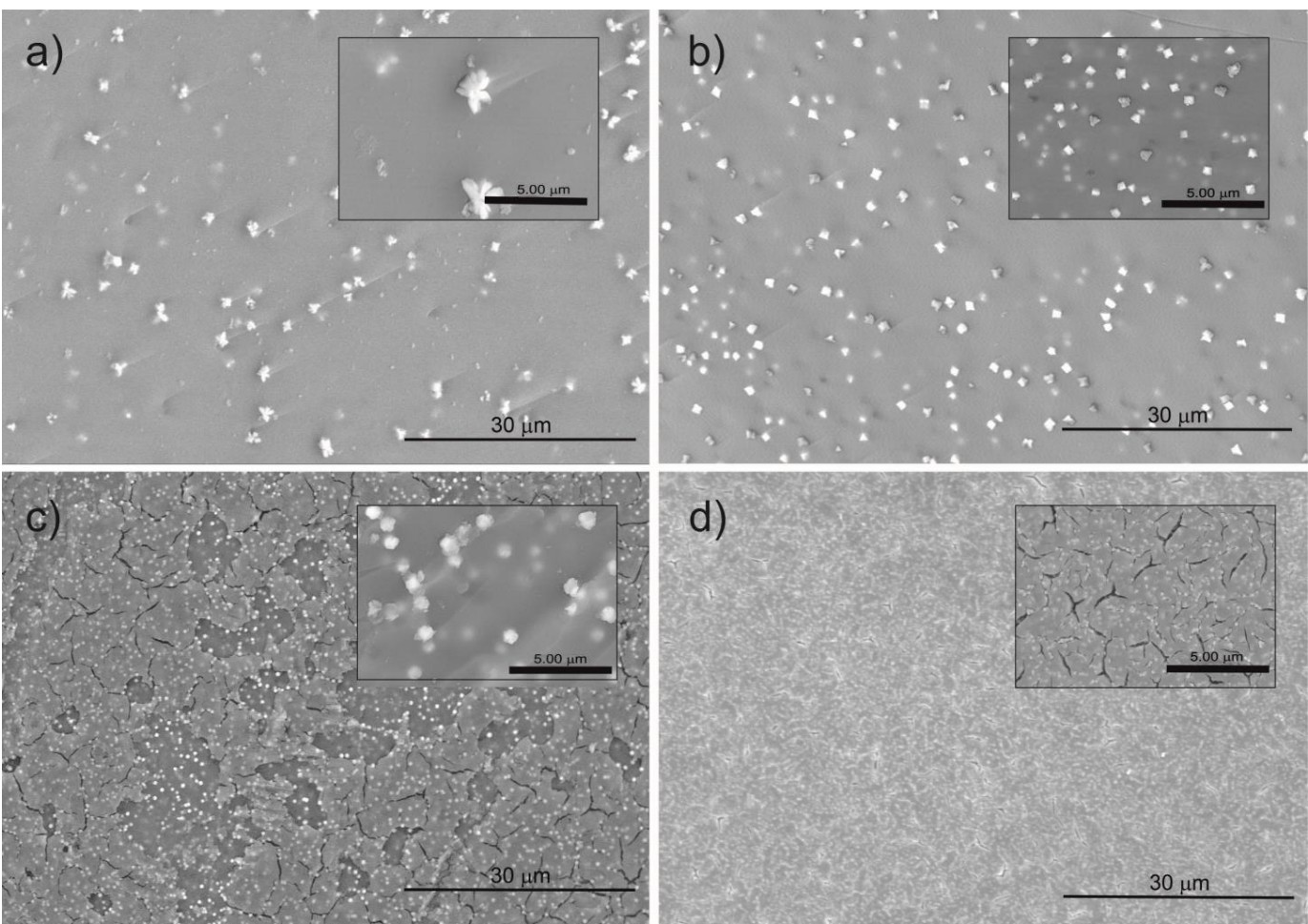

**Figure 9.** Microstructures of the fractures of crystallized glasses in the bulk of the samples. (**a**) AK-0Y. (**b**) AK-3Y. (**c**) AK-5Y. (**d**) AK-12Y.

The micrographs of Figure 9 present the microstructures appearing in the bulk of the crystallized samples. Widespread crystallization is observed all over the bulk sample and the number of crystalline features increases with the $Y_2O_3$ concentration. And just as with surface crystallization, the size of the crystal decreases as the amount of $Y_2O_3$ is raised. Indeed, for the 12% $Y_2O_3$ sample, most of the bulk material is covered with very small crystals. The shape of these crystals also depends on the $Y_2O_3$ content, acquiring a petal-like dendritic shape for samples with no $Y_2O_3$ incorporated whereas, with a small amount of this oxide, the crystals adopt a pyramidal shape. When the concentration of $Y_2O_3$ is then raised to 6 and 12%, the crystals become spherical or globular in shape.

## 4. Discussion

The above results demonstrate the strong influence of the $Y_2O_3$ concentration on the structure and crystallization behavior of multicomponent silicate glasses. From the XRD patterns (Figure 2), FTIR (Figure 3), and Raman (Figure 4) spectroscopies, it is demonstrated that $Y_2O_3$ is incorporated either as network former or modifier depending on its concentration. At low $Y_2O_3$ amounts, this oxide acts as a network former and its role within the glass network changes to glass modifier when its concentration is high enough. For concentrations of 5% and 12%, the Raman spectra show the disappearance of $Q^4$ units ($SiO_2$) to form new $Q^3$ and $Q^2$ units. These results are in accordance with those found from thermal analysis (Figure 4) where the glass stability decreases as the amount of $Y_2O_3$ is increased and the tendency to crystallize increases in the same sense.

Since the main crystallization peaks in the glass materials appeared between 1006 and 1032 °C, the subsequent crystallization process was carried out at 1040 °C. The glasses were heat-treated at this temperature for 30 minutes. It should be noted that there were some remainders in the sample which are still in the glassy state, though. After this crystallization process, the crystalline features developed on both the surface and the bulk of the samples. The XRD patterns of crystallized glasses (Figure 6) revealed the occurrence of different crystalline phases as a function of $Y_2O_3$. If the original glass is produced in the absence of $Y_2O_3$, $ZnSnO_3$ is the unique crystalline feature appearing in the material despite the multicomponent character of the glass sample. Adding just 3% of $Y_2O_3$ to this glass, the $ZnSnO_3$ no longer crystallizes and $ZrSiO_4$ crystallizes instead. According to these pieces of evidence, it can be concluded that $Y_2O_3$ prevents the crystallization of $ZnSnO_3$ and favors the $ZrSiO_4$ crystallization. Increasing the $Y_2O_3$ concentration to 5% leads to the appearance of some other crystalline phases.

The assignment of the diffraction peaks in the XRD patterns to their corresponding reflections allows us to conclude that these minority phases were mainly yttrium silicates (α and β) and $BaSiO_3$. At the maximum amount of $Y_2O_3$, $ZrSiO_4$ remains the majority phase, the reflections of the α- and β-yttrium silicate phases decrease in intensity, and some other reflections attributed to other crystalline phases start to increase. These new crystalline phases that increase, in the sample containing 12% $Y_2O_3$, are calcium silicates (including wollastonite), γ-yttrium silicate, and y-yttrium silicate.

These XRD and Raman spectra results permit us to determine the type of crystals and their evolution during heat treatment as a function of the $Y_2O_3$ concentration in the glass. These crystals are responsible for the white color of the treated samples and can be used as enamels of high reflectivity to minimize the heat transfer into houses to achieve a lower temperature than if such enamels were not used.

Although the above-mentioned crystalline phases appear in these glasses, the presence of two well-defined crystallization regions remains a common feature in all the materials. It is interesting to note the crystallization region starting at the surface of the material which grows inwards independently of the presence or absence of $Y_2O_3$, indicating that the surface crystallization is mainly due to the different oxides of the glass composition. The second crystallization region occurs in the bulk itself. As shown in the micrographs of Figure 8, the thickness of these surface-crystallized zones increases with the $Y_2O_3$ concentration, indicating that this oxide also promotes crystallization at the outer surface of the sample. This phenomenon has also been observed in yttrium aluminosilicate [31, 34, 42] and zinc borosilicate glasses [41]. In turn, as observed in Figure 9, the amount of $Y_2O_3$ also influences the bulk crystallization of the prepared glasses and the concentration of crystals increases while their sizes decrease.

Figures 10 and 11 show the microstructures of the surface and bulk, respectively, as observed at high magnification. As observed in Figure 10, the surface crystallization of the materials containing $Y_2O_3$ can be described with two types of crystallites, with the largest crystals having a long prismatic shape that grows adjacent to the surface. On the other hand, the smallest crystals adopt a pyramidal shape for the 3% $Y_2O_3$ sample and a globular shape in samples AK-5Y and AK-12Y. The size of the small crystals distributed all over the crystallization surface is determined from the SEM micrographs and the averaged data are collected in Table 3. As displayed in Figure 11, the crystals in the microstructure of the bulk material appeared with different shapes depending on the $Y_2O_3$ concentration. In the absence of $Y_2O_3$, the $ZnSnO_3$ crystals are petal-like or dendritic. With incorporation of up to 3% $Y_2O_3$, these petal-like crystals disappear ($ZnSnO_3$ no longer crystallizes) and, instead, star-shaped or pyramidal/cubic-shaped $ZrSiO_4$ crystalline features form. If the $Y_2O_3$ concentration is raised beyond 6%, these small crystals further evolve into a globular/rounded shape. These observations are summarized in Table 3.

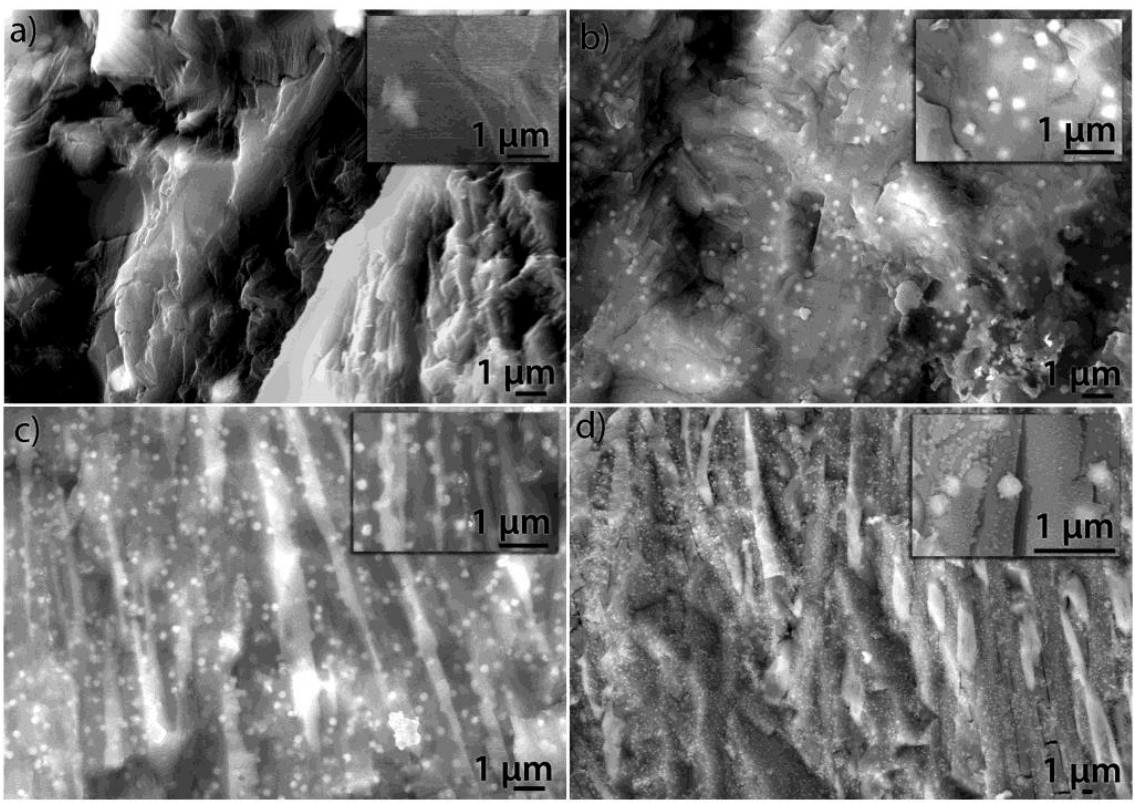

**Figure 10.** Microstructures of the fractures of crystallized glasses at the outer surface of the samples. (**a**) AK-0Y. (**b**) AK-3Y. (**c**) AK-5Y. (**d**) AK-12Y.

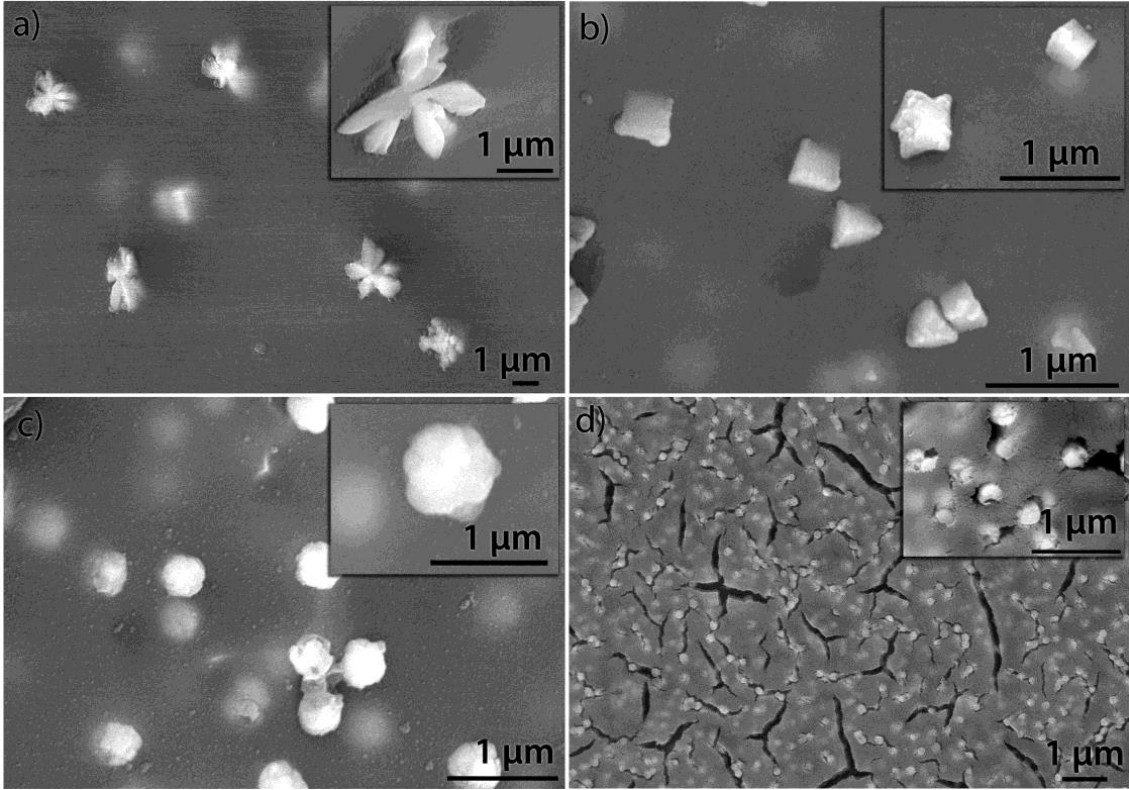

**Figure 11.** Microstructures of the fractures of crystallized glasses in the bulk of the samples. (**a**) AK-0Y. (**b**) AK-3Y. (**c**) AK-5Y. (**d**) AK-12Y.

**Table 3.** Crystal sizes (μm) as observed by FE-SEM analysis (*) or XRD data (‡) and crystallization percentage (*CP*) of the heat-treated glasses determined from FE-SEM images and XRD patterns.

| | AK-0Y | AK-3Y | | AK-5Y | AK-12Y |
|---|---|---|---|---|---|
| Surface | - | 0.40 * | | 0.29 * | 0.19 * |
| Bulk | 1.47 * | 0.53 * | 0.45 * | 0.44 * | 0.25 * |
| Shape | petal-like | polygon | cuboid | globular | globular |
| $ZnSnO_3$ | $0.854 \pm 0.135$ ‡ | - | | - | - |
| $ZrSiO_4$ | | $0.828 \pm 0.054$ ‡ | | $0.804 \pm 0.101$ ‡ | $0.745 \pm 0.130$ ‡ |
| *CP* (%) | 76.70 | 69.77 | | 89.72 | 94.17 |

As a result, it can be concluded that the crystal sizes observed by FE-SEM decrease with the increase in $Y_2O_3$ concentration. This decrease also depends on the shape of the crystals and their location (surface or bulk). The petal-like $ZnSnO_3$ crystals (0% $Y_2O_3$) are slightly larger than the $ZrSiO_4$ ones, and for the case of ZrSiO4 crystals their sizes are larger when they present pyramid/cuboid shapes than for globular crystals (6% and 12% $Y_2O_3$).

XRD data also permit the calculation of the crystallite size by means of the Scherrer equation [43]:

$$L = \frac{K\lambda}{D_{2\theta}\cos\theta} \qquad (4)$$

where $L$ is the crystallite size, $\lambda$ is the X-ray wavelength, $D_{2\theta}$ is the peak width at the half maximum at a particular value of the diffraction angle ($2\theta$), and $K$ is the shape factor, a constant which is a function of the crystallite shape ($K$ = varies from 0.9 to 1 and it is 1 for spherical particles). This equation has been applied to the main peaks of the XRD patterns (Figure 6) and the obtained values of the crystallite sizes are collected in Table 3. In the AK-0Y sample, where only $ZnSnO_3$ is present, the crystallite size is about 0.854 μm. For the other samples, only the $ZrSiO_4$ peaks were considered, although it should be noted that other phases crystallize simultaneously (yttrium silicates, $BaSiO_3$, and wollastonite). Overall, there is a trend of decreasing crystal size as the amount of $Y_2O_3$ is raised.

The XRD patterns in Figure 6 show that the broad glass halo observed in the non-crystallized materials (Figure 2) is still present in the AK-0Y and AK-3Y crystallized samples. However, its intensity decreases with increasing $Y_2O_3$ concentration, indicating a higher degree of crystallization in the samples with more $Y_2O_3$. According to Equation (5), the crystallization percentage (*CP*) can be estimated by:

$$CP = \frac{100 I_c}{\left(I_c + I_g\right)} \qquad (5)$$

In Equation (5), $I_c$ corresponds to the intensity of the main peak of the XRD patterns and $I_g$ is the maximum height of the broad glass halo which is centered at $2\theta$ = 20–40°. The calculated *CP* values are also collected in Table 3. According to these data, it is clear that the heat-treated glasses are highly crystallized especially the sample containing 12% $Y_2O_3$ where more than 94% of the whole material is crystallized. These results are also in accordance with the SEM images where two crystallization zones were observed, indicating that not only is the surface of the material crystallized but so is the bulk. Table 3 reveals that the glass without $Y_2O_3$ incorporated has a higher crystallization degree than the one containing 3% $Y_2O_3$, suggesting in this case that for this low concentration, $Y_2O_3$ acts as a network former of the silicate glass. However, when increasing the $Y_2O_3$ contents to 6% and 12%, the crystallization percentage increases very quickly (69.77% to 94.17%, respectively). Therefore, when $Y_2O_3$ is incorporated in larger amounts, it acts as a nucleating agent promoter of crystallization in these multicomponent glasses. In line with the results reported by Vomacka and Babushkin, small amounts of glassy phase remain in the structure

of the glass-ceramic materials developed with $SiO_2$, $Al_2O_3$, $ZrO_2$, and $Y_2O_3$ compositions because a small amount of $Y_2O_3$ still acts as a network former [22].

Finally, most of the grown crystals have been analyzed by EDS and Figure 12 shows several FE-SEM images with the chemical analysis of different spots carried out in such samples. Table 4 collects the chemical compositions of the most representative crystals.

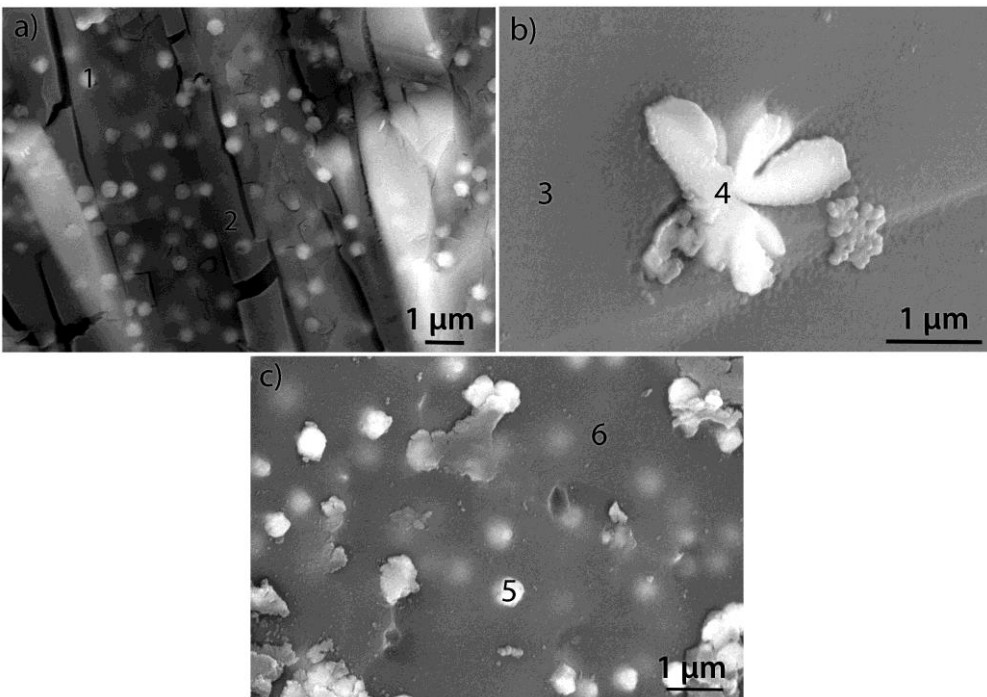

**Figure 12.** EDS analysis carried out in different crystals. (**a**) Representative crystallized surface of all yttrium-containing samples. (**b**) Bulk crystallization in AK-0Y. (**c**) Bulk crystallization in AK-12Y.

**Table 4.** Chemical compositions determined by EDS analysis of the spots shown in Figure 12. Note that points 4 and 5 correspond to the AK-0Y sample.

| Component | 1 | 2 | 3 | 4 | 5 | 6 |
|---|---|---|---|---|---|---|
| $SiO_2$ | 19.07 | 27.27 | 51.69 | 28.35 | 25.23 | 35.20 |
| $Al_2O_3$ | 6.08 | 6.38 | 13.01 | 8.41 | 9.71 | 12.28 |
| $Na_2O$ | 1.03 | 1.21 | 1.02 | 0.08 | 0.82 | 1.25 |
| $K_2O$ | 1.10 | 2.01 | 0.99 | 0.01 | 1.28 | 1.08 |
| MgO | 6.48 | 6.77 | 3.86 | 2.92 | 3.44 | 3.56 |
| CaO | 6.09 | 7.25 | 9.99 | 4.65 | 7.63 | 10.63 |
| BaO | 2.63 | 3.03 | 6.76 | 4.88 | 5.03 | 6.70 |
| ZnO | 7.28 | 7.93 | 4.21 | 12.83 | 3.50 | 5.77 |
| SnO | 7.55 | 1.35 | 1.70 | 33.41 | 11.21 | 1.07 |
| $ZrO_2$ | 8.89 | 9.92 | 6.77 | 4.46 | 11.59 | 3.44 |
| $Y_2O_3$ | 33.80 | 26.98 | 0 | 0 | 20.56 | 19.02 |

The results of Table 4 indicate that the petal-like crystals formed in the AK-0Y sample (points 3 and 4) are mainly formed by SnO and ZnO as concluded by XRD analysis (Figure 6). The other crystals (polygonal or globular) formed on the surface or in the bulk of the samples (points 1 and 5) are mainly composed of zirconium silicate and yttrium

silicates, while those zones on the surface or in the bulk of the samples (points 2, 3, and 6) tend to present a chemical composition close to the original glass before crystallization.

### 5. Conclusions

The effect of different $Y_2O_3$ concentrations (0%, 3%, 5%, and 12%) on the structure and microstructure of multicomponent glasses has been illustrated. XRD, Raman, and FTIR analyses were used for studying the prepared glasses while XRD and FE-SEM were used for the crystallized materials. Raman results revealed a decrease in $Q^4$ units when the $Y_2O_3$ concentration is higher than 3% which is an indication of its network modifier effect. The glass without $Y_2O_3$ crystallizes in $ZnSnO_3$ while those with $Y_2O_3$ crystallize with $ZrSiO_4$ as the main crystalline phase along with different calcium, barium, and yttrium silicates. $Y_2O_3$ prevents the $ZnSnO_3$ crystalline formation in the multicomponent silicate glasses whereas it favors the crystallization of $ZrSiO_4$. For the lowest concentrations of $Y_2O_3$, this oxide promotes the formation of α- and β-yttrium silicates. But for the 12% concentration, the γ- and y-phases of yttrium silicates are also present along with barium and calcium silicates. Despite the crystallization seeming to be a surface-driven process, it occurs either on the surface or in the bulk of the multicomponent glasses. The width of the surface crystallization increases with the $Y_2O_3$ concentration while the concentration of crystals in the bulk also increases with $Y_2O_3$. The microstructures of the crystallized materials are composed of crystals with different sizes and shapes (petal-like, polygonal, globular). These crystals become smaller as the amount of $Y_2O_3$ is raised despite the percentage of crystallized material increasing with higher $Y_2O_3$ content. The chemical compositions determined for the different crystals correspond to the different crystal phases determined by XRD.

**Author Contributions:** Conceptualization, J.R.; methodology, A.B.; software, F.R.; validation, J.R. and A.T.; formal analysis, J.R.; investigation, A.B.; resources, A.C.; data curation, J.R.; writing—original draft preparation, J.R.; writing—review and editing, A.T.; visualization, F.R.; supervision, N.B.; project administration, A.C.; funding acquisition, N.B. and J.R. All authors have read and agreed to the published version of the manuscript.

**Funding:** This research was funded by the Algerian Ministry of Higher Education and Scientific Research (Algerian program P.N.E 2019–2020 scholarship fund) and by the Ministerio de Transición Ecológica of Spain under the project TED2021-132800B-100 financed by the Spanish Research Agency and European Regional Development Fund (AEI/ FEDER, UE). J. Rubio, F. Rubio, and A. Tamayo acknowledge CSIC for the project LincGlobal INCGL20033.

**Data Availability Statement:** The original contributions presented in the study are included in the article, further inquiries can be directed to the corresponding author.

**Acknowledgments:** The authors thank C. Diaz Dorado for the technical assistance with FEM photographs and the design and editing of all figures.

**Conflicts of Interest:** The authors declare no conflicts of interest.

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
