# Peer review of "Effect of Y2O3 Concentration on the Surface and Bulk Crystallization of Multicomponent Silicate Glasses"

_crystals, doi:10.3390/cryst14030214_

Round 1

Reviewer 1 Report

Comments and Suggestions for Authors

Beniaiche et al. presented results on the effect of Y2O3 concentration on the surface and synthesis process of complex silicate glasses. The manuscript is well structured. The research objects are well aligned with the aims and themes of the journal. The manuscript could be published in the journal, although I have a few suggestions.

1. In the Introduction section the authors have clearly and accessibly justified the need for their study, supported by an impressive list of references. Nevertheless, I recommend highlighting the novelty of the study with sufficient clarity as there is a plethora of research in this area.

2. In my opinion, it would be useful to provide the image/images of the samples.

3. Please mark the lines of Raman spectra and DTA curves with more contrast. It is not easy to recognize which line belongs to which sample because the colors appear very similar.

4. DTA graph:

- Indicate the direction of exo- or endo-effects.

- There is an additional peak on the AK-12Y curve (upper one) near the Tm label. Please explain its origin.

Comments on the Quality of English Language

Moderate editing of English language required.

Author Response

Responses to the Reviewer´s questions.

Black colour: Reviewer´s questions/comments

Blue colour: Author´s comments

Red colour: New text/Figure/Table included in the manuscript.

Reviewer 1

Beniaiche et al. presented results on the effect of Y2O3 concentration on the surface and synthesis process of complex silicate glasses. The manuscript is well structured. The research objects are well aligned with the aims and themes of the journal. The manuscript could be published in the journal, although I have a few suggestions.

  1. In the Introduction section the authors have clearly and accessibly justified the need for their study, supported by an impressive list of references. Nevertheless, I recommend highlighting the novelty of the study with sufficient clarity as there is a plethora of research in this area.

Response: We have included the novelty of this study as Referees recommend. The included new text is:

The effect of ZrO2 and ZrSiO4 has been widely studied for their incorporation in multicomponent glasses and glass enamels in the ceramic tile sector, however this effect has not been analyzed for the Y2O3 even this oxide gives interesting properties related to their high refraction index and the possibility to produce glass enamels of high whiteness.

  1. In my opinion, it would be useful to provide the image/images of the samples

Response: A new Figure has been included in the Section 2.1 of the manuscript. Then all the numbers of the old Figures have been changed. The included new text is (in Section 2.1):

Figure 1. Images of the as-prepared (up-side) and crystallized (down-side) glasses.

Figure 1 shows an image of the as-prepared and crystallized glasses and it can be observed the white color obtained after crystallization due to the formation of different crystals as it will be commented below.

  1. Please mark the lines of Raman spectra and DTA curves with more contrast. It is not easy to recognize which line belongs to which sample because the colors appear very similar.

Response: Thank you for your appreciation. More contrast colors have been used and besides the positions of the main Raman bands are added to the spectra. We think this can help to follow the discussion of the spectra.

  1. DTA graph:

- Indicate the direction of exo- or endo-effects.

- There is an additional peak on the AK-12Y curve (upper one) near the Tm label. Please explain its origin.

Response: Thank you for this observation.  The additional endo peak has been included and discussed in the text. It has been also included the direction of exo-endo effects. Then, the included new text is:

In can be observed how for the AK-12Y sample there appear two endothermic Tm peaks which indicate at least two phases that melt at very close temperatures. As it will be commented below this glass gives several crystalline phases that can present different melting temperatures.

Besides, all equations, Figure 5 and results of Table 2 have been changed or modified in accordance with the Reviewer´s comment.

Reviewer 2 Report

Comments and Suggestions for Authors

It is an interesting article and can be of interest to readers.

1- Highlight the aspect of novelty research in the introduction.

2- How do the results of XRD and Raman spectra analysis affect the efficiency of the samples? You should discuss this in depth.

3- EDX-MAPPING elemental analysis should be added to the article.

4- Besides glass, what other substrates can be used? What is the advantage of glass over quartz and other substrates?

5- State other uses of synthesized samples in the introduction.

6- Is it possible to Synthesize several layers of these materials?

7- Add a schematic of the synthesis to the article.

8- I suggest you optimize some parameters such as pH and concentration in the synthesis.

Comments on the Quality of English Language

Minor editing of English language required

Author Response

Responses to the Reviewer´s questions.

Black colour: Reviewer´s questions/comments

Blue colour: Author´s comments

Red colour: New text/Figure/Table included in the manuscript.

Reviewer 2

It is an interesting article and can be of interest to readers.

  • Highlight the aspect of novelty research in the introduction.

Response: This is a similar question as Referee 1 commented and then, the include new text has been:

The effect of ZrO2 and ZrSiO4 has been widely studied for their incorporation in multicomponent glasses and glass enamels in the ceramic tile sector, however this effect has not been analyzed for the Y2O3 even this oxide gives interesting properties related to their high refraction index and the possibility to produce glass enamels of high whiteness.

  • How do the results of XRD and Raman spectra analysis affect the efficiency of the samples? You should discuss this in depth.

Response: Thank you for your observation and a new paragraph has been included in the 4.Discussion Section. This is:

These XRD and Raman spectra results permit to determine the type of crystals and their evolution during heat treatment as function of the Y2O3 concentration in the glass. These crystals are responsible of the white color of the treated samples and they can be used as enamels of high reflectivity to minimize the heat transfer into the houses to achieve a lower temperature than if such enamels were not used.

  • EDX-MAPPING elemental analysis should be added to the article.

Response: Thank you again for this comment. We had analysed all the samples but we had not include the obtained results to not increase the length of the paper. We agree with Reviewer´s comment in that with these analyses the paper is improved. However, after a careful study of the results we think that is better to carried out a spot EDS analysis rather than an EDS-mapping because the wide composition of these glasses do not give high differences between different points. Then, in the new text we have included the more representative EDS analysis based on different spots on crystals and in the bulks of the samples. All this is now added to the paper, and it is as follows:

Finally, most of the grown crystals have been analyzed by EDS and Figure 12 shows several FE-SEM images with the chemical analysis of different spots carried out in such samples. Table 4 collects the chemical compositions of the most representative crystals.

Figure 12. EDS analysis carried out in different crystals. a) Representative crystallized surface of all Yttrium containing samples. b) Bulk crystallization on AK-0Y. c) Bulk crystallization in AK-12Y.

Table 4. Chemical compositions determined by EDS analysis of the spots showed in Figure 12. Note that points 4 and 5 correspond to the AK-0Y sample.

Component

1

2

3

4

5

6

SiO2

19.07

27.27

51.69

28.35

25.23

35.20

Al2O3

6.08

6.38

13.01

8.41

9.71

12.28

Na2O

1.03

1.21

1.02

0.08

0.82

1.25

K2O

1.10

2.01

0.99

0.01

1.28

1.08

MgO

6.48

6.77

3.86

2.92

3.44

3.56

CaO

6.09

7.25

9.99

4.65

7.63

10.63

BaO

2.63

3.03

6.76

4.88

5.03

6.70

ZnO

7.28

7.93

4.21

12.83

3.50

5.77

SnO

7.55

1.35

1.70

33.41

11.21

1.07

ZrO2

8.89

9.92

6.77

4.46

11.59

3.44

Y2O3

33.80

26.98

0

0

20.56

19.02

Results of Table 4 indicate that the petal-like crystals formed in the AK-0Y sample (points 3 and 4) are mainly formed by SnO and ZnO as it was concluded by XRD analysis (Figure 6). The other crystals (polygon or globular) formed on the surface or in the bulk of the samples (points 1 and 5) are mainly formed by Zirconium silicate and Yttrium silicates, while those zones on the surface or in the bulk of the samples (points 2, 3 and 6) tend to present a chemical composition close to the original glass before crystallization.

  • Besides glass, what other substrates can be used? What is the advantage of glass over quartz and other substrates?

Response: Quartz can be also used and as it is described Experimental Section (2.1 Glass preparation and crystallization) it is the base of the glass composition because its concentration is between 38-50% as it is reported in Table 1 (Section 2.2. Characterization). The main problem of the quartz is its low refractive index (about 1.54-1.56) so it is necessary to incorporate other oxides of high refractive indexes as ZrO2 or Y2O3. Because ZrO2 is widely studied, the objective of this work is to study the effect of Y2O3 because it has only been studied for glasses with two or three different oxides.

5- State other uses of synthesized samples in the introduction.

Response: Thank you again for this observation. The new text included in the Introduction section is:

These crystallized materials can be used as glass enamels for high reflecting surfaces in ceramic tiles for roof and façades, home ovens, fillers for white paints, etc.

6- Is it possible to synthesize several layers of these materials?

Response: Yes, it is possible but the crystallization mechanism should be studied again because it could be possible that the heat diffusion affect such mechanism for each layer. This is an important question to be studied in a great detail and, we thank to the reviewer to think us about this new possibility. This is another work to be carried out, however, this is not commented in the reviewed version.

7- Add a schematic of the synthesis to the article.

8- I suggest you optimize some parameters such as pH and concentration in the synthesis.

Response to questions 7 and 8: In this case we don´t know what to comment because we don´t use any solution (nor organic nor aqueous) for the material preparation. We only mix raw inorganic materials that are then melted at high temperatures and cooled down to solid state and finally heat treated for obtaining the corresponding crystals inside the glass structure.  Therefore, if we add a schematic of the synthesis it will be only four boxes with: mixing, melting, cooling and treating. And all of this is already written in the Section 2.

Reviewer 3 Report

Comments and Suggestions for Authors

The authors prepared multicomponent silicate glass crystals, studied the effect of different concentrations of Y2O3 on the structure of multicomponent glasses, and characterized and analyzed the multicomponent silicate glass crystals using XRD, Raman, FTIR and FE-SEM. This manuscript can be accepted after addressing the following problems:

Format needs to be modified:

1) The expression of the exothermic peak of crystallization in the paper is inconsistent with that in Figure 4. (e.g. The text is TP while the figure shows TC).

2) The decimal point in the ordinate in Figure 4 is used incorrectly.

3) Whether to use italics for physical quantities, and the format should be uniform. (e.g. The format of Tg, TX and Tm are different from KH).

4) The numbering colors on the image are inconsistent. (e.g. figure 7, figure 9).

5) Equation 5 is not on the same line.

6) The format of the references is not uniform, authors need to pay attention to details. (e.g. Abbreviated Journal Name).

Pictures can be improved in detail:

   The feature bands (special bands described in the article) above Figures 2, 3, and 6 can be indicated with arrows or other signs to make it easier for readers to understand.

Line 145, it should be “Y2O3

Line 199, Whether γ-Y2Si2O7 is β-Y2Si2O7? Because samples containing 3% and 5% Y2O3 were not observed γ-Y2Si2O7 crystallization in figure 5. In addition, the author also discuss later on γ-Y2Si2O7 is a new crystalline phase that appears in samples containing 12% Y2O3. The author needs to examine this problem.

Line 368 says that the calculated CC values are also collected in Table 3, but the CC column is not seen in Table 3.

Author Response

Responses to the Reviewer´s questions.

Black colour: Reviewer´s questions/comments

Blue colour: Author´s comments

Red colour: New text/Figure/Table included in the manuscript.

Reviewer 3

The authors prepared multicomponent silicate glass crystals, studied the effect of different concentrations of Y2O3 on the structure of multicomponent glasses, and characterized and analyzed the multicomponent silicate glass crystals using XRD, Raman, FTIR and FE-SEM. This manuscript can be accepted after addressing the following problems:

Format needs to be modified:

  • The expression of the exothermic peak of crystallization in the paper is inconsistent with that in Figure 4. (e.g. The text is TPwhile the figure shows TC).

Thank you. This has been corrected in both the Figure and Text.

  • The decimal point in the ordinate in Figure 4 is used incorrectly.

Thank you. This has been corrected in both the Figure.

  • Whether to use italics for physical quantities, and the format should be uniform. (e.g. The format of Tg, TXand Tm are different from KH).

Response: Thank you again. We have corrected the text and all physical quantities are not italics, and the equations are now in italics.

4) The numbering colors on the image are inconsistent. (e.g. figure 7, figure 9).

Response: Thank you again. It has been corrected in all the Figures.

5) Equation 5 is not on the same line.

Response: Thank you, it has been corrected.

6) The format of the references is not uniform; authors need to pay attention to details. (e.g. Abbreviated Journal Name).

 Response: Thank you; all references have been corrected if necessary.

Pictures can be improved in detail:

  The feature bands (special bands described in the article) above Figures 2, 3, and 6 can be indicated with arrows or other signs to make it easier for readers to understand.

Response: Thank you again. Yes, we agree with Referee and all the positions of the main bands are indicated in the Figures except for the FTIR spectra because there only appear three bands clearly recognized.

Line 145, it should be “Y2O3

Response: Thank you, it has been correct.

Line 199, Whether γ-Y2Si2O7 is β-Y2Si2O7? Because samples containing 3% and 5% Y2O3 were not observed γ-Y2Si2O7 crystallization in figure 5. In addition, the author also discuss later on γ-Y2Si2O7 is a new crystalline phase that appears in samples containing 12% Y2O3. The author needs to examine this problem.

            Response: Thank you very much for your appreciation and we found that the text was not in agreement with our results and what we want to explain. When we had written that the phases a-Y2Si2O7 and g-Y2Si2O7 "disappear" in the AK-12Y sample, we should have written "decrease" and the y-Y2Si2O7 phase appears acquiring a higher intensity than both a and g yttrium silicate phases. The corrected text is now as:

A close analysis to the XRD patterns reveals that both the a-Y2Si2O7 and g-Y2Si2O7 crystalline phases are still present in the samples containing 3% and 5% Y2O3 and these phases decrease in the AK-12Y sample and the y-Y2Si2O7 crystallizes instead acquiring a higher intensity than both a and g Yttrium silicate phases.

This has been also reviewed in the 4 Discussion Section, as:

These new crystalline phases that increase, in the sample containing 12% Y2O3, are calcium silicates (including wollastonite), g-yttrium silicate and y-yttrium silicate.

Line 368 says that the calculated CC values are also collected in Table 3, but the CC column is not seen in Table 3.

Response: Thank you again. It has been correct.

Round 2

Reviewer 2 Report

Comments and Suggestions for Authors

Accept in present form

Comments on the Quality of English Language

 Minor editing of English language required